# Kinase Inhibition in Multiple Myeloma: Current Scenario and Clinical Perspectives

**DOI:** 10.3390/pharmaceutics14091784

**Published:** 2022-08-25

**Authors:** Igor Valentim Barreto, Caio Bezerra Machado, Davi Benevides Almeida, Flávia Melo Cunha de Pinho Pessoa, Renan Brito Gadelha, Laudreísa da Costa Pantoja, Deivide de Sousa Oliveira, Rodrigo Monteiro Ribeiro, Germison Silva Lopes, Manoel Odorico de Moraes Filho, Maria Elisabete Amaral de Moraes, André Salim Khayat, Edivaldo Herculano Correa de Oliveira, Caroline Aquino Moreira-Nunes

**Affiliations:** 1Pharmacogenetics Laboratory, Department of Medicine, Drug Research and Development Center (NPDM), Federal University of Ceará, Fortaleza 60430-275, CE, Brazil; 2Unichristus University Center, Faculty of Biomedicine, Fortaleza 60430-275, CE, Brazil; 3Department of Biological Sciences, Oncology Research Center, Federal University of Pará, Belém 66073-005, PA, Brazil; 4Department of Hematology, Fortaleza General Hospital (HGF), Fortaleza 60150-160, CE, Brazil; 5Department of Hematology, César Cals General Hospital, Fortaleza 60015-152, CE, Brazil; 6Faculty of Natural Sciences, Institute of Exact and Natural Sciences, Federal University of Pará (UFPA), Rua Augusto Correa, 01, Belém 66075-990, PA, Brazil; 7Laboratory of Cytogenomics and Environmental Mutagenesis, Environment Section (SAMAM), Evandro Chagas Institute (IEC), BR 316, KM 7, s/n, Levilândia, Ananindeua 67030-000, PA, Brazil; 8Northeast Biotechnology Network (RENORBIO), Itaperi Campus, Ceará State University, Fortaleza 60740-903, CE, Brazil

**Keywords:** multiple myeloma, tyrosine kinase, treatment, ibrutinib

## Abstract

Multiple myeloma (MM) is a blood cell neoplasm characterized by excessive production of malignant monoclonal plasma cells (activated B lymphocytes) by the bone marrow, which end up synthesizing antibodies or antibody fragments, called M proteins, in excess. The accumulation of this production, both cells themselves and of the immunoglobulins, causes a series of problems for the patient, of a systemic and local nature, such as blood hyperviscosity, renal failure, anemia, bone lesions, and infections due to compromised immunity. MM is the third most common hematological neoplasm, constituting 1% of all cancer cases, and is a disease that is difficult to treat, still being considered an incurable disease. The treatments currently available cannot cure the patient, but only extend their lifespan, and the main and most effective alternative is autologous hematopoietic stem cell transplantation, but not every patient is eligible, often due to age and pre-existing comorbidities. In this context, the search for new therapies that can bring better results to patients is of utmost importance. Protein tyrosine kinases (PTKs) are involved in several biological processes, such as cell growth regulation and proliferation, thus, mutations that affect their functionality can have a great impact on crucial molecular pathways in the cells, leading to tumorigenesis. In the past couple of decades, the use of small-molecule inhibitors, which include tyrosine kinase inhibitors (TKIs), has been a hallmark in the treatment of hematological malignancies, and MM patients may also benefit from TKI-based treatment strategies. In this review, we seek to understand the applicability of TKIs used in MM clinical trials in the last 10 years.

## 1. Introduction

Multiple myeloma (MM) is a systemic hematological neoplasm in which there is an abnormal proliferation of malignant monoclonal plasmocytes that release antibodies or antibody fragments, called M proteins, in excess [1,2,3,4,5]. This feature is responsible for a range of symptoms of MM such as blood hyperviscosity and damage to the renal tubules. Along with this accumulation, the interaction of malignant plasma cells with other cells in the bone marrow (BM) causes several problems for patients such as anemia, destructive bone lesions, and infections due to compromised immunity [5,6,7].

MM is the third most reported neoplasm representing approximately 1% of all cancers and approximately 10% of all hematological malignancies, with risk factors for MM development involving gender, age, family history of malignancies, and ethnicity [1,8,9]. MM is more commonly reported in men than in women with a ratio of 1.5 to 1 and being twice as common in African Americans compared to Caucasians [2,10,11]. The median age at diagnosis tends to be around 65 years [2,12].

MM emerges from molecular changes caused by DNA damage and failures in DNA repair mechanisms [13]. As a genetically complex disease, MM development is a process composed of several stages, with initial mutations appearing in hematopoietic stem cells (HSC) of the bone marrow (BM), the site most affected by the disease [14]. With the onset of malignant transformations, patients enter a pre-malignant stage called smoldering multiple myeloma (SMM) or monoclonal gammopathy of undetermined significance (MGUS) that occurs due to genetic events such as chromosomal translocations involving immunoglobulin heavy chain *(IgH)* genes and aneuploidy [15]. Secondary genetic events, such as copy number abnormalities and acquired mutations, are linked to tumor progression [16,17,18,19]. With the accumulation of mutations that guarantee competitive advantages, HSCs evolve into malignant cells and begin to proliferate causing accumulation of malignant plasma cells in the BM [19]. This proliferation is sustained also due to the release of cytokines, such as interleukin 6 (IL-6), carried out by BM stromal cells [5,20,21]. The genetic alterations found in the MGUS stage are involved in tumor development, while the events present in the MM stages that were not found in MGUS are responsible for tumor progression [12,17].

MGUS is an asymptomatic pre-malignant phase, preceding most cases of MM and being present in approximately 3–4% of the population over 50 years of age. Of the total number of MGUS cases, only 1% per year has a chance of progression to MM [22,23,24,25]. Patients with translocation t(4;14), del(17p), gain(1q), and trisomies have a higher risk of progression from the SMM stage to MM, reaching 10% per year. This being also a risk factor for progression from MGUS to MM [26,27,28].

Regarding genetic alterations, MM is known to present a great heterogeneity; however, mutations that could be considered precursors of the disease are now well known, and these genetic alterations can be used as a prognostic factor. An example of this would be mutations in the *IgH* gene locus, responsible for producing the heavy chains of immunoglobulins, which are considered to be early precursor mutations [3,29,30]. Chromosomal translocations involving *IgH* and other genes such as *Nuclear Receptor Binding SET Domain Protein* (*NSD2*), *F**ibroblast Growth Factor Receptor 3* (*FGFR3*), *Cyclin D3* (*CCND3*), *Cyclin D1* (*CCND1*), *MAF bZIP Transcription Factor* (*MAF*), and *MAF bZIP Transcription Factor B* (*MAFB*), resulting in the translocations t(4;14), t(6;14), t(11;14), t(14;16), and t(14;20), respectively, are also an important clinical finding, as they deregulate checkpoints of the cell cycle due to increased gene transcription under the activity of *IgH* transcription enhancer (Figure 1) [14,31,32,33,34].

## 2. Current Clinical MM Treatment Options

Overall survival of MM cases has more than doubled in recent decades due to the introduction of new combinations of chemotherapy, small molecule inhibitors, and the use of monoclonal antibodies [35,36,37,38].

Hematopoietic stem cell transplantation (HSCT) is still the most recommended treatment for MM, being the first choice in most cases, although not all patients are eligible. Non-eligibility can happen for a variety of reasons, including age, which is an important cut-off point for inclusion criteria, pre-existing comorbidities, and performance score, which is used to predict poor outcomes in patients with MM [39,40,41].

When eligible for HSCT, patients are submitted to one of two induction regimens: VTD, which includes bortezomib, thalidomide, and dexamethasone, or VRD, which includes bortezomib, lenalidomide, and dexamethasone, being the most adopted pre-transplant induction regimens available [42]. The role of induction chemotherapy is to reduce the neoplastic burden at the patient’s BM in order to increase response rates and effectiveness of an autologous transplantation graft [43,44,45,46]. Satisfactory response rates can be seen in young, transplant-eligible patients receiving high-dose melphalan therapy with autologous stem cell transplantation achieving a >60% effective response [47].

Most patients afflicted with MM end up relapsing and those who relapse and are not eligible for a new HSCT require a triple therapy regimen that varies from case to case. Although highly cytotoxic, triple therapy regimens should be continued until the toxicity outweighs the benefits or until patients are eligible for autologous HSCT [35,38]

On the other hand, the use of proteasome inhibitors revolutionized the management of hematological malignancies emerging as one of the most important agents for the treatment of MM [48]. Tumor cells are proteasome-dependent to eliminate excess proteins that arise due to the continuous production of monoclonal immunoglobulin chains. Proteasome hyperactivity in MM results in the degradation of important proteins such as the nuclear factor kappa B (NFκB), the enzymatic complex of inhibitors of nuclear factor kappa B (IκB), tumor protein p53 (TP53) suppressor proteins, among other proteins responsible for the cell cycle [49,50]. Its inhibition leads to cellular stress induced by high protein load in the endoplasmic reticulum due to the accumulation of intracellular proteins, leading to cell death in MM cells [51,52]. Three agents in this class are approved by the Food and Drug Administration (FDA) for use, bortezomib, carfilzomib, and ixazomib [53,54,55,56,57].

Approved in 2003, bortezomib was the first proteasome inhibitor to be used for the treatment of relapsed and refractory multiple myeloma. Reversibly binding with high affinity to the 20S proteolytic core withing the 26S proteasome without inhibiting other types of proteases commonly present in the human body, bortezomib inhibits the ubiquitin–proteasome pathway, triggering a series of events such as induction of apoptosis, cell cycle inhibition, angiogenesis and adhesion, and cell proliferation. Despite being highly potent and effective, bortezomib has a limitation during treatment: the dose that will be used is limited by the toxicity of the drug that is often associated with peripheral neuropathy [52,55,56,58,59].

Carfilzomib is a second-generation proteasome inhibitor approved in 2012 for use in monotherapy or doublet or triplet combination regimens, especially for patients with relapsed or refractory multiple myeloma. Unlike bortezomib, which binds reversibly, carfilzomib binds irreversibly and highly selectively to the 20S proteasome, precisely in the chymotrypsin-like β5 subunit. The inhibition of this subunit is enough to cause apoptosis due to the accumulation of proteasome substrates inside the cell, which could explain the high sensitivity of MM cells to this drug, use of carfilzomib being possible in a monotherapy regimen aiming to reduce adverse effects. In this context, the drug has lower neurotoxicity compared to bortezomib, but there is a higher possibility of causing hypertension, congestive heart failure, and coronary artery disease, not being recommended for patients with heart diseases [54,55,56].

Ixazomib was approved in 2015 for use in combination with lenalidomide and dexamethasone (Rd) in patients who have received at least one therapy regimen previously. It is the first proteasome inhibitor that can be administered orally. Ixazomib is a potent and selective inhibitor of 20S proteasome, also binding in the chymotrypsin-like β5 subunit, as carfilzomib, but reversibly, as bortezomib. In addition, at high concentrations, the drug can also bind in two other subunits of this proteasome: β1 caspase-like and β2 trypsin-like, increasing the selectivity of ixazomib by proteasome 20S. Ixazomib has no cardiotoxic effects, but neurotoxicity is present, although at a lower level when compared to bortezomib. The most reported adverse effects are thrombocytopenia, skin rash, and GI symptoms—diarrhea, nausea, and vomiting [54,55,60,61,62].

Another important therapy in the treatment of MM is the use of immunomodulators (IMiDs), such as thalidomide, pomalidomide, and lenalidomide. In general, immunomodulators act by inhibiting cell growth by inducing apoptosis in MM cells through the inhibition of interferon regulatory factor 4 (IRF4), thus, affecting expression of the *MYC* proto-oncogene (MYC). The overexpression of these two factors is linked to oncogenesis in several types of cancer, including MM, as they are involved in the processes of regulation of cell growth and metabolism, differentiation, apoptosis, angiogenesis, DNA repair, protein translation, and hematopoietic cell formation. In MM, they act by regulating the immune response and the development of immune cells [63,64,65,66].

In addition, IMiDs act favoring the production of interleukin-2 (IL-2) and interferon-gamma (IFN-γ), activating T lymphocytes and natural-killer (NK) cells, on the other hand also inhibiting the production of the tumor necrosis (TNF)-α. The discovery of the inhibitory effects of thalidomide on tumor progression led to the development of two other analogues, lenalidomide and pomalidomide, which were authorized and released for use in MM after clinical trials in 2006 and 2013 [64,67,68,69].

Daratumumab is a new kind of drug that has been used since 2015 for the treatment of MM and it has been promoting an incredible improvement in the efficiency of the treatment [70]. Daratumumab is an anti-CD38 monoclonal antibody, which is a pleiotropic glycoprotein highly expressed on plasma cells and MM cells, acting as a transmembrane receptor on these cells [70,71]. The role of CD38 as a receptor involves signaling for cell activation and proliferation and inducing cell adhesion processes [72], which may explain its high expression in MM tumor cells. Therefore, the daratumumab’s monoclonal antibodies directly target and destroy tumor cells due to several mechanisms, including antibody-dependent cell-mediated cytotoxicity, complement-dependent cytotoxicity, antibody-dependent phagocytosis, and immune cell depletion or inhibition of immunosuppressive cells, constituting a specific type of immunotherapy for patients with multiple myeloma [46,73,74,75,76]. Initially, it was only used as a monotherapy regimen for patients with relapsed or refractory MM, but its low toxicity allowed it to be added to other drugs in several different combinations, both in triple and quadruple therapies [70,71].

Emerging therapies with better outcomes involve immunotherapy and personalized medicine based on the molecular characteristics of the patient’s tumor [33,71,77]. Specific antibodies against B-cell maturation antigen (BCMA) have shown promise in the treatment of MM, since it is an antigen whose expression is higher in myeloma cells than in healthy plasma cells, having an essential role in the process of maturation and differentiation of B-cells, being only expressed on antibody-producing B lymphocytes [78,79]. Antibodies with dual specificity are also under development, the central idea of which is to bind a T cell to a tumor cell, with the antibody acting as a binding bridge, and thus induce the destruction of the tumor cell by the T lymphocyte connected to it [71,77].

Due to the evolution of molecular profiling techniques, it is possible to identify several genetic and molecular abnormalities that constitute the neoplastic clones of a specific patient. The knowledge of the existing mutations helps guide the best available treatment based on the genetic characteristics of each patient, in addition to assessing the patient’s prognosis, thus, determining whether a more aggressive drug therapy is necessary or if the patient is in a situation of good prognosis and may be eligible for autologous HSCT [31,33,80].

With the understanding that MM is a disease still treated with a non-curative approach, the constant development of new therapeutic strategies is one of the main goals in oncologic investigations and routine clinical practice. In the past couple of decades, the use of small-molecule inhibitors, which include tyrosine kinase inhibitors (TKI), has been a hallmark in the treatment of hematological malignancies, and MM patients may also benefit from TKI-based treatment strategies [31,32].

## 3. Tyrosine Kinase Inhibitors in MM

Protein tyrosine kinases (PTKs) are part of a large, multigene family and their main functions are to coordinate cellular behavior, regulate mitosis, differentiation, apoptosis, and a series of physiological and biochemical processes [81,82,83]. Structurally, PTKs can be divided into receptor PTKs (RTK), acting as receptors for external signals of growth and survival factors and phosphorylating other protein residues in the intracellular compartment, and non-receptor PTKs (NRTK), which are cytoplasmic or nuclear proteins that act as second messengers. Examples of both classes include insulin-like growth factor 1 receptor (IGFR), mast/stem cell growth factor receptor Kit (KIT), hepatocyte growth factor receptor (MET), fibroblast growth factor receptor (FGFR3), vascular endothelial growth factor receptor (VEGFR), and platelet derived growth factor receptor (PDGFR), as RKTs, and Bruton’s tyrosine kinase (BTK), Janus kinase (JAK), SRC proto-oncogene (SRC), ABL proto-oncogene (ABL), and FA complementation group (FAC), as NRTKs [81,84,85,86,87].

Several groups of diseases present alterations linked to PTK, as their abnormal expression is linked to disorders in the regulation of cell proliferation, leading to the process of tumorigenesis, and their overexpression is also related to invasion and metastasis, tumor neovascularization, and resistance to chemotherapy [81,85,86,88,89].

There are currently 71 TKIs approved by the FDA for the treatment of neoplasms (http://www.brimr.org/PKI/PKIs.htm, accessed on 21 July 2022). Acquired resistance remains a problem in cancer-targeted therapies as a variety of resistance mechanisms are described in TKI treatment protocols, such as amplification of target receptor expression, mutations in tyrosine kinase inhibitor binding receptors, overactivation of alternative cell survival pathways, and activation of downstream signaling effectors linked to cell proliferation [82,90]

Table 1 is composed of a series of clinical trials over the last 10 years using TKIs as monotherapy or in combination with other cytotoxic agents to treat patients afflicted with refractory multiple myeloma and their results with degrees of efficacy.

Of the 11 articles described in Table 1, 36.4% (4) addressed the treatment with ibrutinib as a major option for MM. The other studies addressed treatments with other kinase inhibitors, such as Ruxolitinib, Cabozantinib, INCB052793, Tivantinib, Trametinib, Afuresertibe, Sorafenib, and Dovitinib. In total, 36.4% (4) of the articles described in the table are phase I clinical trials, while the other 63.6% (7) are phase II clinical trials [92,93,94,95,96,97,98,99,100,101].

## 4. Ibrutinib: A BTK Inhibition Approach

BTK inhibitors (BTKi) are one of the most popular and advanced approaches to targeting the BCR pathway. In addition to having revolutionized the treatment of B-lymphocyte malignancy, they also have a high level of efficacy in relation to chronic lymphocytic leukemia (CLL) patients, especially those with high-risk mutations [102,103]. Among these inhibitors, Ibrutinib stands out, already showing robust and durable efficacy in the treatment of refractory CLL and being one of the first inhibitors of the pathway to be approved by the FDA [104,105].

Ibrutinib is an FDA-approved drug for the treatment of B-cell malignancies [106], it works by irreversibly binding BTK through a covalent bond with a cysteine residue at position 481 (C481) [107]. It has demonstrated clinical responses mainly related to refractory CLL and mantle cell lymphoma (MCL), but it also has approval for use in cases of Waldenström’s macroglobulinemia, small lymphocytic lymphoma, and marginal zone lymphoma. Chronic inhibition of BTK has been shown to be so effective in terms of its anticancer activity that its drugs are being widely tested in hematologic malignancies and solid malignancies [108,109]. Since overexpression of BTK is present in 85% of MM cases, ibrutinib appears as a promising therapy for MM patients, and the roles of BTK in the development of bone resorption by osteoclasts, as well as in cell migration, are characteristics that support BTK’s research in the context of MM development [108,110,111,112,113].

BTK belongs to the Tec tyrosine kinase family that is involved in the B-cell antigen receptor (BCR) signaling pathway, being related to the survival, proliferation, and progression of malignancies in these cells. In MM, BTK is related to drug resistance, bone disease, and increased cell proliferation [114]. BTK activity, together with tyrosine phosphorylation, triggers the action of protein kinase B (AKT) which in turn will mediate transcription factors for proliferation, differentiation, and signaling cascades for survival—RAS/RAF/MEK/ERK and PI3K/AKT/mTOR (Figure 2) [110,114,115,116,117].

In the four studies presented in Table 1 in which ibrutinib was used, satisfactory results were reported, reaching conclusions where the clinical response to the use of ibrutinib encourages its use. This can also be seen in the statistical data of these articles, such as ORR above 60% [91,92], clinical response of 57% [93], and CBR of 28% [94]. The number of patients presented in the studies totaled 268, with an average age above 60 years. The preferred dose of ibrutinib was 840 mg, which is the limit dose for patients with MM, in addition to being a higher dose than that used in the treatment of other diseases such as lymphoma and chronic lymphocytic leukemia (CLL), which already have approved doses of 560 and 420 mg, respectively. It is worth mentioning that patients with CLL do not usually use doses of 840 mg due to the risk of discontinuation caused by adverse effects (AEs) [118].

All four studies [91,92,93,94] utilized drugs which are standards for MM therapies in combination with ibrutinib, varying between carfilzomib and bortozomib, but the association of ibrutinib with dexamethasone was unanimous among all groups. The association of the BTK inhibitor with other drugs seeks to potentiate the therapeutic action and increase rates such as overall response rates (ORR) and progression-free survival (PFS) [94]. This association is also observed in cases of CLL, since BTKis in monotherapy are not enough to obtain profound responses and are therefore used in combination with other drugs to increase efficiency and not lose their effects due to resistance mechanisms [103,105]. In the study carried out by Richardson et al. [94], the combination of ibrutinib with dexamethasone demonstrated better results when compared to ibrutinib monotherapy. Chari et al. [91] observed that the combination of ibrutinib with carfilzomib and dexamethasone showed a promising response.

In studies using ibrutinib, among the AEs suffered by patients, the presence of anemia, diarrhea, and thrombocytopenia was constant [91,92,93,94]. AEs related to BTKi inhibitors are still being described, however, cardiac effects are the main concern due to the risk of combining these with hemorrhagic effects [108]. In in vitro studies, ibrutinib has already been demonstrated to interact with collagen-dependent platelet activation and von Willebrand factor, and it has been linked to an increased incidence of ventricular arrhythmia, hypertension, and neutropenia [110,111,118,119].

Dickerson et al. [54] observe that 78.3% of the patients who were using ibrutinib developed or worsened hypertension in an average period of 30 months. Although the causal relationship between ibrutinib and cardiotoxicity has not yet been fully elucidated, there are some theories, one of which would be the ability of ibrutinib to bind with other Tec kinases [120]. Despite cases of cardiotoxicity such as atrial fibrillation (AF), as long as the patient is benefiting from the therapy, they may continue with the treatment, with follow-up and medication to control the possible AEs. Next-generation BTK inhibitors, with less capacity to cause cardiotoxicity, are being studied and considered for MM treatment [120,121,122].

## 5. Clinical Perspectives with Other TKIs

In general, the use of tyrosine kinase inhibitors alone does not present a satisfactory response in patients with refractory MM [123]. The use of sorafenib alone in 7 out of 11 patients did not stop progression of MM, presenting a PFS of 2.6 months [100], while the isolated use of tivantinib was shown to be well tolerated, but in a clinical trial with 11 patients the agent was only able to stabilize 4 of 11 (36%) patients with progressive myeloma [124].

Other studies show that isolated administration of trametinib and afuresertib presented a significant clinical improvement in patients with MM, but when they are associated with other medications patients have even better responses [125,126]. Since trametinib is a drug also used to treat melanoma, its mechanism is based on MEK inhibition [127] and afuresertib is an AKT inhibitor that can also be used in the treatment of ovarian cancer [128]. However, the association between these two kinase inhibitors for the treatment of refractory MM patients performed in the study by Tolcher et al. [99] did not show such promising results, because despite having obtained maximum tolerated dose (MTD) values, these were compatible with subtherapeutic doses, thus, rending the maintenance of the drugs in clinically significant concentrations for a long period of time impossible due to the AEs.

In the study carried out by Scheid et al. [101], the activity of dovitinib, which acts on the fibroblast growth factor receptor 3 (*FGFR3*), was evaluated in MM patients. *FGFR3* is an RTK of the FGFR family that is responsible for cell growth, differentiation, and migration in a wide variety of cell types and is present in MM and a variety of cancers. Upon ligand stimulation, a dimerization of the receptor occurs followed by transphosphorylation of tyrosine residues in the intracellular domain signaling mainly through extracellular signal-regulated kinase (ERK) 1 and 2 pathways, PI3K and PLC [129,130,131,132,133,134].

Better results from dovitinib were observed in patients with MM with the presence of t (4;14). This translocation is associated with a worse prognosis, causing the overexpression of the *MMSET* (multiple myeloma SET domain protein) and *FGFR3* genes [135,136]. Thus, the interaction between genetic abnormalities can be further explored, bringing new insights into personalized therapy.

It was possible to perceive a wide variety of occurrences in the AEs manifested by the patients participating in the analyzed studies. In general, the most recurrent AEs were diarrhea, nausea, fatigue, dermal toxicity, anemia, leukopenia, thrombocytopenia, infections, hypertension, and congestive heart failure [95,96,97,98,99,100].

Due to many clinical difficulties in the treatment of refractory MM, it is important to point out that the search for newer effective therapies extends also to the promising use of TKIs, which demonstrated the positive ability to prevent the progression of the disease in several studies described here. However, it is worth emphasizing the importance of follow-up studies and clinical trials with different drug combinations to obtain better clinical results and avoiding AEs [137].

A new alternative to this would be other BTK inhibitors that are already approved by the FDA and even those in the study phase. Ibrutinib, despite its satisfactory results, presents the problem, already discussed, regarding the AE profile and its susceptibility to resistance pathways. However, since its mechanism of action demonstrates benefits, drugs that act in a similar way appear as a good proposal for future research and treatment. An example would be zanubrutinib and acalabrutinib, which are already approved by the FDA, and despite having irreversible links in their sites of action, are able to be more selective and show fewer problems related to platelet dysfunction and bleeding. Another proposal that has been showing good prospects is the non-covalent, reversible BTK inhibitors, that, among their advantages, have a greater selectivity for the site of action and are also effective in patients with resistance to ibrutinib [138,139,140,141].

## 6. Conclusions

In this review, we observed that among the TKIs tested in the last 10 years for the treatment of refractory MM, Ibrutinib was the most used and presented better clinical results, manly when administered in association with other drugs to avoid the emergence of resistance mechanisms that have already been found in other hematological neoplasms. Although AEs emerging from TKI’s clinical administration is a major problem, when properly addressed and managed, treatment-emergent AEs are not considered serious, and the patient benefit versus risk ratio must be measured and taken into account individually from case to case.

## Figures and Tables

**Figure 1 pharmaceutics-14-01784-f001:**
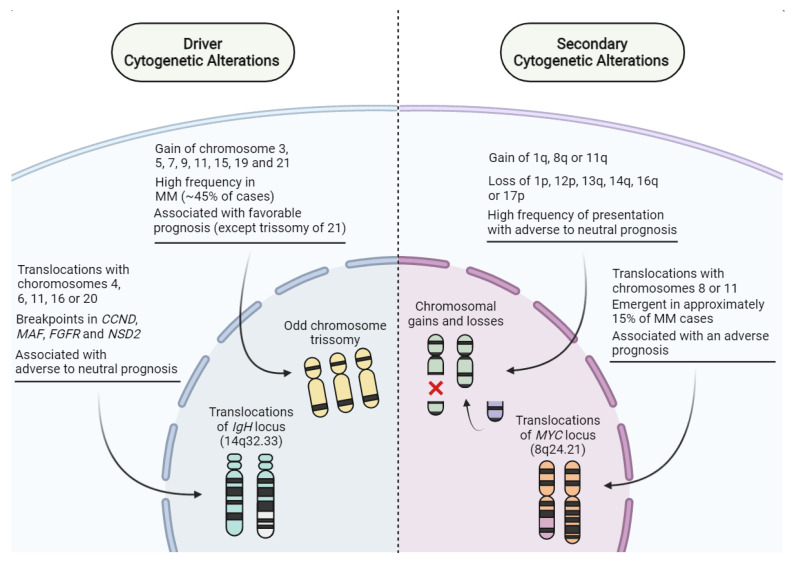
Cytogenetic alterations associated with multiple myeloma (MM). In pre-malignant settings, driver alterations correlate with deregulation of oncogene activity and carcinogenesis onset. At later disease stages, secondary alterations emerge due to genomic instability in malignant clones, and MM cells from the same patient may even harbor different secondary alterations, following the concepts of linear and branching evolution. Created with BioRender.com.

**Figure 2 pharmaceutics-14-01784-f002:**
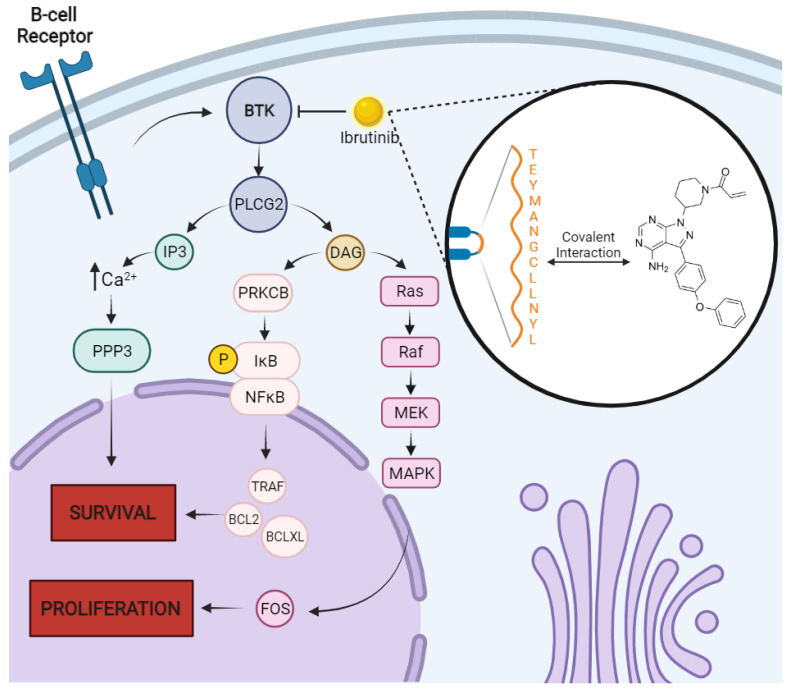
Bruton’s tyrosine kinase (BTK) survival and proliferation pathways and interaction with ibrutinib. In multiple myeloma (MM), B-cell receptors of malignant plasma cells signal for BTK cascade to begin, which is highly overexpressed and induces survival and proliferation of neoplastic clones. Downstream activity happens mainly through phospholipases which catalyze formation of IP3 (inositol 1-4-5 trisphosphate) and DAG (diacylglycerol), being effectors of intracellular calcium signaling pathways, in the case of IP3, and NFκB and MAPK signaling pathways, in the case of DAG. Inhibition of BTK by ibrutinib, however, can drastically disrupt the metabolism of MM cells and happens through the covalent and irreversible interaction of ibrutinib with the cysteine residue on position 481 of the BTK active domain. Created with BioRender.com.

**Table 1 pharmaceutics-14-01784-t001:** TKIs used in clinical trials to treat refractory multiple myeloma in the last 10 years.

Clinical Study Phase	Targeted Kinase	Kinase Inhibitor	Associated Treatment	Clinical Outcome	Adverse Events	References
I	BTK	Ibrutinib	Carfilzomib/Dex-methasone	ORR of 67% and a PFS of 7.2 months.	Hypertension, anemia, pneumonia, fatigue, diarrhea, and thrombocytopenia.	[91]
I/IIb	BTK	Ibrutinib	Carfilzomib/Dexamethasone	Acceptable safety profiles with PFS and ORR of 7.7 months and 71%, respectively. The average one-year OS rate was 77%.	Thrombocytopenia, anemia, diarrhea, fatigue, nausea, and hypertension.	[92]
II	BTK	Ibrutinib	Bortezomib/Dexamethasone	The drug combination initially increased the levels of infections and risk minimization measures were necessary. Clinical response was observed in 57% of patients with a duration of 9.5 months.	Thrombocytopenia, diarrhea, anemia, asthenia, and pneumonia.	[93]
II	BTK	Ibrutinib	Dexamethasone	The highest CBR was achieved in the combination of ibrutinib 840 mg with dexamethasone 40 mg. With CBR of 28%, ORR of 5%, sustained SD of 23%, and median PFS of 4.6 months.	Diarrhea, fatigue, nausea, anemia, and thrombocytopenia.	[94]
I	JAK	Ruxolitinib	Lenalidomide and metilprednisolone	The drug showed the ability to abrogate resistance to lenalidomide. Featuring CBR of 46% and ORR of 38%.	Anemia, thrombocytopenia and lymphopenia, sepsis, and pneumonia.	[95]
Ib	HGF and MET	Cabozantinib	NR	The drug alone has no significant activity in patients with refractory MM. The study was interrupted, and the rates were not calculated.	Grade 2 congestive heart failure and grade 3 APN. The remaining AEs were related to intestinal events.	[96]
Ib	JAK1	INCB052793	NR	No significant responses were observed. ORR of 24% and OS of 6.7 months.	Thrombocytopenia, anemia, fatigue, nausea, and vomiting.	[97]
II	c-MET	Tivantinib	NR	In isolation, the drug did not present a satisfactory response in refractory patients. SD of 36% and PD of 63% were obtained.	Neutropenia, hypertension, syncope, infection, and pain.	[98]
II	MEK and AKT	Trametinib and Afuresertib	NR	MTDs were found, being concentrations that are below the monotherapy concentration of each drug. However, these doses were considered subtherapeutic.	Diarrhea, acneiform dermatitis, maculopapular rash, fatigue, dry skin, nausea, dyspnea, and vomiting.	[99]
II	VEGF	Sorafenib	NR	Only one patient completed the 13 cycles of treatment and achieved PR, another 7 patients remained in PD.	Fatigue, nausea, hypertension, dermal toxicity, hematologic toxicity, and heart attack.	[100]
II	FGFR3	Dovitinib	NR	The SD rate in t(4;14)-positive patients was higher, being 61.5%, compared with 34.6% rates for those translocation-negative	Diarrhea, nausea, vomiting, and fatigue.	[101]

Legend: NR: not reported; SD: stable disease; PR: partial response; ORR: overall response rate; PD: patients showed progression; OS: median overall survival; PFS: progression-free survival; CBR: clinical benefit rate; MTD: maximum tolerated dose.

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
