# Peer review of "Kinase Inhibition in Multiple Myeloma: Current Scenario and Clinical Perspectives"

_pharmaceutics, 2022, doi:10.3390/pharmaceutics14091784_

Round 1

Reviewer 1 Report

To the Authors:

The manuscript submitted aims to review the applicability of tyrosine kinase inhibitors (TKIs) used in the context of MM clinical trials in the past 10 years.

Abstract: The abstract reflects the content of the manuscript. The authors state that «MM is the second most common hematological neoplasm, constituting 1% of all cancer cases» but according to the latest update by the IARC (WHO) it is currently the third most common. The same correction applies to the Introduction.

Keywords: The keywords are appropriate.

Introduction: The introduction is consistent with the subject.

The last 2 paragraphs appear somewhat out of sequence. The epidemiological considerations regarding MM/ MGUS could make more sense if placed in the beginning of the Introduction. Also reference 24 (Rajkumar, S.V. Multiple Myeloma: 2020 Update on Diagnosis, Risk-stratification and Management. American Journal of Hematology 2020, 95, 548–567, doi:10.1002/ajh.25791.) has a 2022 update (Am J Hematol.2022;97:1086–1107The).

The phrase « Due to its asymptomatic nature, progression from MGUS to MM is related to the rate of 1% per year» is confusing, leading the reader to the assumption that it is because it is asymptomatic that the progression rate is 1%. Similarly, the statements « Due to the genetic disorder caused by MM, such as the gain of t(4;14), del(17p) and gain(1q) translocations, it is related to an accelerated progression from SMM to MM, being about 10% per year. This is a risk also associated with accelerated progression from MGUS to MM» should be clarified.

Suggested corrections: Smoldering instead of latent MM; Fibroblast Growth Factor Receptor 3 (FGFR3) instead of FGFR; Cyclin D3 (CCND3) instead of CCND.

Current Clinical Treatment Options: Something seems to be missing for the first statement to make sense (maybe MM treatment options?).

The phrase« Satisfactory response rates can be seen in young, transplant-eligible… response» as it refers to response to a particular treatment regimen would be better placed when that treatment is developed further on in the manuscript.

Again, statements regarding the treatment regiments should consider updated information (as previously suggested).

Although the authors refer to 3 proteasome inhibitors only 1 is discussed.

The phrase «The discovery of the inhibitory effects of thalidomide on tumor progression was made with the development of two other analogues…» should be clarified. Do the authors mean that it led to the development of analogues?

The phrase «Therefore, the Daratumumab’s monoclonal antibodies directly target and destroy tumor cells…» should be clarified.

The paragraph regarding molecular profiling, although adding relevant information, has no clear association with the previous or the following statements.

Suggested  corrections: response rates instead of responses rates; immune cells instead of immune system cells

Tyrosine Kinase Inhibitors in MM: The statement «Changes linked to PTK are related to a diverse group of diseases, more than 50% of proto-oncogenic and oncogenic products present abnormalities in PTK activities» could be better phrased.

Suggested corrections: Dexamethasone instead of Dexametasona; Metilprednisolone instead of metilprednisolona; the abbreviation MTD is not explained; Clinical Benefit Rate should be CBR instead of CRB as it is sometimes used (Table 1).

Ibrutinib: What do the authors mean by «cell proliferation disorders»?

What do the authors mean by «traditional chemistry»?

Suggested corrections: increase overall response rates instead of increase results values. AEs seen in other pathologies instead of AEs seen in other pathologies also treated with them.

Clinical Perspectives with others TKIs:  The abbreviation MDT is not explained.

Suggested corrections: well tolerated, and in a clinical trial instead of well tolerated, in a clinical trial; MDT values, these were compatible instead of MDT values, their values; Rending impossible instead of existing the impossibility; described instead of describe.

I would recommend that the manuscript undergoes an English revision, particularly, the use of very long sentences (ex. Conclusion section).

Author Response

Dear reviewer, my co-authors and I would like to thank you for the suggestions made during this high-quality review and then we present the answers to the questions.

We inform that with the reviews and suggestions, we were able to improve the idea presented by our work and we appreciate the opportunity. We hope this review has left the article suitable for publication in this high-impact journal and respect in the area.

Kind Regards.

Response to reviewer 1

The manuscript submitted aims to review the applicability of tyrosine kinase inhibitors (TKIs) used in the context of MM clinical trials in the past 10 years. 

Abstract: The abstract reflects the content of the manuscript. The authors state that «MM is the second most common hematological neoplasm, constituting 1% of all cancer cases» but according to the latest update by the IARC (WHO) it is currently the third most common. The same correction applies to the Introduction.

R= Corrections have been accepted and information has been updated.

Keywords: The keywords are appropriate.

Introduction: The introduction is consistent with the subject. 

The last 2 paragraphs appear somewhat out of sequence. The epidemiological considerations regarding MM/ MGUS could make more sense if placed in the beginning of the Introduction. Also reference 24 (Rajkumar, S.V. Multiple Myeloma: 2020 Update on Diagnosis, Risk-stratification and Management. American Journal of Hematology 2020, 95, 548–567, doi:10.1002/ajh.25791.) has a 2022 update (Am J Hematol.2022;97:1086–1107The). 

R= The paragraphs have been reallocated and the reference updated.

The phrase « Due to its asymptomatic nature, progression from MGUS to MM is related to the rate of 1% per year» is confusing, leading the reader to the assumption that it is because it is asymptomatic that the progression rate is 1%. Similarly, the statements « Due to the genetic disorder caused by MM, such as the gain of t(4;14), del(17p) and gain(1q) translocations, it is related to an accelerated progression from SMM to MM, being about 10% per year. This is a risk also associated with accelerated progression from MGUS to MM» should be clarified. 

R= The sentence has been rewritten for clarity.

Suggested corrections: Smoldering instead of latent MM; Fibroblast Growth Factor Receptor 3 (FGFR3) instead of FGFR; Cyclin D3 (CCND3) instead of CCND.

R= Corrections were carried out as suggested.

Current Clinical Treatment Options: Something seems to be missing for the first statement to make sense (maybe MM treatment options?). 

R= The phrase has been replaced by "Current Clinical MM Treatment Options".

The phrase« Satisfactory response rates can be seen in young, transplant-eligible… response» as it refers to response to a particular treatment regimen would be better placed when that treatment is developed further on in the manuscript. 

Again, statements regarding the treatment regiments should consider updated information (as previously suggested). 

R= The phrase has been relocated as suggested.

Although the authors refer to 3 proteasome inhibitors only 1 is discussed. 

R= Two new paragraphs have been added discussing the other proteasome inhibitors.

The phrase «The discovery of the inhibitory effects of thalidomide on tumor progression was made with the development of two other analogues…» should be clarified. Do the authors mean that it led to the development of analogues? 

R= The sentence has been rewritten for clarity.

The phrase «Therefore, the Daratumumab’s monoclonal antibodies directly target and destroy tumor cells…» should be clarified. 

R= The sentence has been rewritten for clarity.

The paragraph regarding molecular profiling, although adding relevant information, has no clear association with the previous or the following statements. 

R= Due to the relevance of the information, we tried to rewrite part of the paragraph for better clarification.

Suggested corrections: response rates instead of responses rates; immune cells instead of immune system cells

R= Suggestion accepted and corrected.

Tyrosine Kinase Inhibitors in MM: The statement «Changes linked to PTK are related to a diverse group of diseases, more than 50% of proto-oncogenic and oncogenic products present abnormalities in PTK activities» could be better phrased. 

R= The sentence has been rewritten for clarity.

Suggested corrections: Dexamethasone instead of Dexametasona; Metilprednisolone instead of metilprednisolona; the abbreviation MTD is not explained; Clinical Benefit Rate should be CBR instead of CRB as it is sometimes used (Table 1).

R= Corrections made and the meaning of MTD was specified in the legend of table 1 and in the text.

Ibrutinib: What do the authors mean by «cell proliferation disorders»? 

R= In the text the sentence was replaced by "increased cell proliferation".

What do the authors mean by «traditional chemistry»? 

R= We refer to traditional chemotherapy, this correction was made in the text.

Suggested corrections: increase overall response rates instead of increase results values. AEs seen in other pathologies instead of AEs seen in other pathologies also treated with them. 

R= Suggestion accepted and corrected.

Clinical Perspectives with others TKIs:  The abbreviation MDT is not explained. 

R= The meaning of MTD was specified in the legend of table 1 and in the text.

Suggested corrections: well tolerated, and in a clinical trial instead of well tolerated, in a clinical trial; MDT values, these were compatible instead of MDT values, their values; Rending impossible instead of existing the impossibility; described instead of describe. 

 R= Suggestion accepted and corrected.

I would recommend that the manuscript undergoes an English revision, particularly, the use of very long sentences (ex. Conclusion section).

R= The article underwent a new English revision.

Reviewer 2 Report

The review is an up-to-date overview of the state of the art of tyrosine kinase inhibitors used in multiple myeloma clinical trials over the past 10 years. The introduction is concise and the discussion soon focuses on the results. The review is easy enough to read and requires moderate revision of English. In my opinion, however, the review lacks innovation. The authors report on the TKIs tested in the last 10 years for the treatment of refractory MM, focusing on Ibrutinib, which, besides being the most widely used, has presented better clinical results. However, the authors could also make a comparison with potential new tyrosine kinase inhibitors in the literature to make this review more complete.

After these improvements, the work could be published in a journal of the quality of Pharmaceutics

Author Response

Dear reviewer, my co-authors and I would like to thank you for the suggestions made during this high-quality review and then we present the answers to the questions.

We inform that with the reviews and suggestions, we were able to improve the idea presented by our work and we appreciate the opportunity. We hope this review has left the article suitable for publication in this high-impact journal and respect in the area.

Kind Regards.

Response to reviewer 2

The review is an up-to-date overview of the state of the art of tyrosine kinase inhibitors used in multiple myeloma clinical trials over the past 10 years. The introduction is concise and the discussion soon focuses on the results. The review is easy enough to read and requires moderate revision of English. In my opinion, however, the review lacks innovation. The authors report on the TKIs tested in the last 10 years for the treatment of refractory MM, focusing on Ibrutinib, which, besides being the most widely used, has presented better clinical results. However, the authors could also make a comparison with potential new tyrosine kinase inhibitors in the literature to make this review more complete.

After these improvements, the work could be published in a journal of the quality of Pharmaceutics

R= A paragraph was added to the end of the topic "Clinical Perspectives with other TKIs" addressing the new treatment perspectives.
